

# Multivariate analysis of extreme storm surges in a semi-enclosed bay

## Yao Luo[1], Hui Shi[2], Dongxiao Wang[1*]

[1]State Key Laboratory of Tropical Oceanography, South China Sea Institute of Oceanology, Chinese Academy of Sciences, Guangzhou, China

[2]China Water Resources Pearl River Planning Surveying & Designing Co., Ltd., China

*Corresponding author: Dongxiao Wang (dxwang@scsio.ac.cn)

**Abstract:**

The prediction of extreme storm surges is a critical task for coastal area protection. This study examines extreme storm surges in Beibu Bay, a semi-enclosed bay in the South China Sea, and their joint probabilities. A method for the advanced prediction of the extreme storm surges is proposed using a multivariate extreme statistical method. We further present practical guidelines of the proposed multivariate analysis method, including guidelines for simulation. The simulation can be

extended to multidimensional data to simplify computation, so the proposed approach can be extended to use more points' data from the semi-enclosed bay for predicting extreme storm surges probabilities. A practical case study illustrates the application of the proposed techniques for extreme storm surges prediction. A comparison of the conditional probabilities obtained from observations and simulation data show that the proposed method is effective.

**Key Words:** Multivariate extreme analysis, Monte Carlo, SCS, Storm Surge

## 1. Introduction

The prediction of extreme storm surges due to typhoons is a critical problem for low-lying coastal areas. This problem is expected to increase because of climate change (Lowe et al., 2010; Weisse et al., 2012; Weisse et al., 2014; Mcinnes et al., 2003; Tebaldi et al., 2012; Arns et al., 2015).

Inundation caused by storm surges in the Chinese coastal region severely impacts and poses a continual threat to the activities and safety of the people living there. Between 1990 and 2010, storm surge disasters caused economic losses of approximately RMB 10.5 billion and 148 deaths, and affected 11.5 million annually. (Gao et al., 2014). Thus, storm surge prediction and rapid disaster





information dissemination are vital for facilitating evacuation and disaster mitigation in China.

This study focuses on extreme storm-surge probability prediction in Beibu Bay in the South China Sea (SCS) using a multivariate extreme statistical method called multivariate generalized Pareto distribution (MGPD). Recently, research on the application of multivariate extreme values (EVs) has increased. In these studies, several possible probability distribution functions have been used to characterize extreme sea level events: the Copula function (Salvadori et al., 2015; Corbella

and Stretch, 2012; Michele et al., 2007; Salvadori et al., 2013; Tao et al., 2013), multivariate EV function (Morton and Bowers, 1996; Coles and Tawn, 1994; Bhunya, et al., 2011), and MGPD function (which is a type of multivariate EV function) (Falk et al., 2004; Rootzén and Tajvidi, 2006).

In fact, multivariate EV analysis requires a reasonable extrapolation from observed states to unobserved states. Therefore, in EV analysis, these distribution functions are not only fitted by

observed states, but they also predict unobserved states by reasonable extrapolation. The multivariate EV and MGPD functions are derived from EV theory, a branch of probability theory (Coles, 2001; Beirlant et al., 2005). However, only the MGPD function is the natural distribution of the multivariate peak over threshold (MPOT) sampling method, which retains more extreme information from the raw data than the annual maxima method (Zhang et al., 2000; Yap and Zhu,

2014; Menéndez and Woodworth,2010). The annual maxima method often ignores multiple severe storm waves that occur in the same year, which may be much larger than the annual largest waves in many other years. Consequently, the annual maxima method may underestimate extreme variables (You and Yin, 2006). Moreover, the uncertainty of the return level estimation by the peak over threshold method is smaller than that by the annual maxima method under different sample

lengths (Yao et al., 2012).

In this paper, we do not specifically address MGPD theory; its details can be found in (Falk, 2004; Rootzén and Tajvidi, 2006; Beirlant et al., 2005; Tajvidi, 1996). We instead present our developed MGPD procedures and show how the methodology can be exploited for the analysis of extreme surges at two adjacent sites. The proposed theory and its statistical methodology are

presented in Section 2. In MGPD, determining the joint threshold and estimating the joint density are critical tasks. These aspects are presented via an example in Sections 3 and 4, respectively.



Finally, MPOT advantages and application possibilities based on Monte Carlo simulation are outlined in the conclusion in Section 5.

## 2. Analysis Methodology

### 2.1 MGPD Theory

Multivariate EV theory has a broad range of applications (Coles and Tawn, 1994; Beirlant et al., 2005; Coles and Tawn, 1991). In this paper, we focus on the MGPD definition of Falk et al. (2004). The MGPD cumulative distribution function can be described as

$$W(X) = 1 + \log(G(x_1, ..., x_d))$$

$$= 1 + \left( \sum_{i=1}^{d} x_i \right) D \left( \frac{x_1}{\sum_{i=1}^{d} x_i}, ..., \frac{x_{d-1}}{\sum_{i=1}^{d} x_i} \right), \qquad \log\left(G\left(x_1, ..., x_d\right)\right) > -1 \qquad (1)$$

where $(x_1, ..., x_d) = x \in U$, where $U$ is a neighborhood of zeros in the negative quadrant $(-\infty, 0)^d$, and $G(x_1, \cdots, x_n)$ is a cumulative distribution function of the multivariate generalized EV distribution. In addition, $D$ is the Pickands dependence function in the unit simplex $\overline{R_{d-1}}$ on the defined domain, i.e., $\overline{R_d} = \{x \in [0, \infty)^d \mid \sum_{i=1}^{d} x_i = 1\}$.

Function $W(X)$ can be deduced from $G(x_1, \cdots, x_n)$, so the dependence relation form of $G(x_1, \cdots, x_n)$ can greatly enrich the expression of W(X) for various dependence relationships (Falk et al., 2004). Among them, $W(X)$ of the logistic type is widely used in hydrology, finance, and other fields because of the favorable statistical properties of the following Pickands dependence function:

$$D_r\left(t_1, ..., t_{d-1}\right) = \left( \sum_{i=1}^{d-1} t_i^r + \left( 1 - \sum_{i=1}^{d-1} t_i \right)^r \right)^{1/r}, \qquad (2)$$

which means the MGPD cumulative distribution function can be rewritten as

$$W_r(x) = 1 - \left( \sum_{i=1}^{d} (-x_i)^r \right)^{1/r} = 1 - \|x\|_r, \qquad (3)$$

where $r$ is the correlation parameter of the dependence function and $r > 1$. Further, $x_i$ in the interval



$(-1, 0)$ denotes standardized variables. The bivariate logistic generalized Pareto distribution density function is given by

$$w_r(x, y) = \frac{\partial W}{\partial x \partial y} = (r-1)(xy)^{r-1}[(-x)^r + (-y)^r]^{1/r-2} \qquad x < 0, y < 0. \tag{4}$$

Correlation parameter $r$ can be evaluated using a step-by-step method, i.e., we first evaluate two marginal distributions and then evaluate $r$ within MGPD including the marginal distributions. Alternatively, $r$ can be evaluated using a global method in which $r$ is directly estimated using the maximum likelihood for density function $w$. The global method more reliably evaluates the results because of its final function form; however, its evaluation processes are more complex. The

maximum-likelihood function is

$$L(r) = \sum_{i=1}^{n} \ln(w_r(x_i, y_i)). \tag{5}$$

### 2.2 Simulation method

  The Monte Carlo simulation method for a multivariate distribution is complex owing to the generation of multivariate random vectors. In general, the variables firstly are transformed into

independent variables, for which random vectors are generated for each variable. The final random vectors of the multivariate distribution are obtained by an inverse transformation. The MGPD simulation method was suggested in Michel (2007).

  To more effectively demonstrate the MGPD simulation method, we use $T_p$ to transform vector $(x_1, ..., x_d)$ into polar coordinates as follows:

$$T_p(x_1, ..., x_d) = (\frac{x_1}{x_1 + ... + x_d}, ..., \frac{x_{d-1}}{x_1 + ... + x_d}, x_1 + ... + x_d) = (z_1, ..., z_{d-1}, c), \tag{6}$$

where $c = x_1 + \cdots x_d$ and $\mathbf{Z} = (x_1/c, \cdots, x_{d-1}/c)$ correspond to the radial and angular components, respectively, called Pickands polar coordinates.

  In the Pickands polar coordinate system, $W(X)$ represents different properties. Assume that $(X_1, ..., X_d)$ follows MGPD $W(X)$ and the $d$-th differential of its Pickands dependence

function $D$ exists. We define the Pickands density $\phi(z, c)$ as follows (Falk et al., 2004):





$$\phi(z,c) = |c|^{d-1}\left(\frac{\partial^d}{\partial x_1, \cdots, \partial x_d} W\right) T_p^{-1}(z,c) \tag{7}$$

Then $\phi$ depends only on $z$ and, therefore, we put $\phi(z) = \phi(z,c)$. Under

$\mu = \int_{R_{d-1}} \phi(z)dz > 0$ and constant $c_0 < 0$ exists and is close to zero, we can prove that c is a

uniform distribution P (C > c) = μ|c| on $(c_0, 0)$. For details see Falk et al. (2004). So MGPD

simulation method (details in Michel, 2007): 1) generates random numbers uniformly distributed on

unit simplex $\overline{R_{d-1}}$, 2) generates random vector $(z_1, \cdots z_d)$ using density function $f(z) = \frac{\phi(z)}{\mu}$ of

$z = (z_1, \cdots, z_{d-1})$ in Pickands polar coordinates and an acceptance-rejection method, 3) generates

random numbers uniformly distributed on $(c_0, 0)$, and 4) calculates vector

$\left(cz_1, \cdots, cz_{d-1}, c - c\sum_{i=1}^{d-1} z_i\right)$, which is a random vector that satisfies the multivariate over

threshold distribution. Constant $c_0$ is the joint threshold in the MGPD method, determined in this

study using the principle proposed in (Coles and Tawn, 1994).

## 3. Data and Declustering

### 3.1 Data

The data used in this study were provided by the Joint Archive for Sea Level (JASL) of the

University of Hawaii Sea Level Center (http://uhslc.soest.hawaii.edu/home). The data consist of

simultaneous hourly sea-level observations at Beihai and Dongfang stations, which are located on

the Beibu Bay coast in SCS (Fig. 1) and were collected from June 1975 to December 1997. The data

can be used for the analysis of extreme surges because hourly sampling sufficiently captures high

water levels. The missing values for Beihai correspond to only 0.023% of the data, while for

Dongfang, 0.173% of the data was eliminated from the sample because of gauge malfunctions or

other issues. 201,578 and 201,275 hourly values for Beihai and Dongfang, respectively, were to be

processed, and the available data was judged sufficient to perform EV analysis of storm surges.

### 3.2 Data Analysis and Declustering

Beihai City, located in the south of Guangxi province of China, is a beach city on the coast of




Beibu Bay, which is a shallow, semi-closed bay. Owing to Beibu Bay's special geomorphologies,

its typhoon surges are violent and can cause flooding in the city. The surge levels at the site are

defined as the residuals after the astronomically induced tidal components have been removed from

the sea-level observations. The tidal component is cyclical and does not satisfy the basic hypothesis

of a random variable. Godin's method (Godin, 1972) was used for tidal analysis.

The first stage in EV analysis is declustering, specifically, identifying a set of independent

events. Declustering is performed to make adjacent elements of a sample that consists of the maxima

of events independent of each other. s. Our approach is to use a moving window of sufficient width

to ensure that the extremes of each variable from a single 'meteorological event' are certain to fall

within a single window (Coles and Tawn, 1994). Declustering techniques by Morton and Bowers

(1996) were used. The features of storms and storm surges differ in each place. In Beibu Bay, the

main meteorological event, can generate extreme wave height, is a typhoon. The declustering of a

sequence of surges is illustrated in Fig. 2. The duration of a typhoon surge in the Beibu Gulf is

approximately 100 h. The components of each vector are the maximum surge at each site over a

100-h event. The peak events for the vectors in the cluster occurred at different times. The peak

surges arrived at Dongfang 3-5 h before they arrived at Beihai, as shown in Fig. 2.

The main purpose of this study was to analyze the relationship between the extreme surges at

both sites, and then predict extreme storm surge probabilities at one site. Therefore, the declustering

method was judged to be appropriate because all the main peaks were included in these clusters.

The 100-h cluster interval enabled the surge peaks at both sites from the same typhoon to be in the

same cluster. According to the above principles, the total number of independent events $N$ was *2,016*.

### 3.3 Constructing Conditional Probability Functions

Extreme surges in the Beibu Gulf is are predominantly caused by typhoons from lower-latitude

areas in the SCS. Typhoons typically move through the Beibu Gulf from south to north, with a small

number of them moving from east to west. An extreme surge at Dongfang, which is to the south of

Beihai, can serve as an early warning signal for Beihai. Multivariate EV analysis can be used to

provide such a warning.

To analyze the joint probability of extreme surges in Beihai and Dongfang, conditional





probability (*CP*) distributions can be used (eq. 9). The value of *CP* represents the probability of

encounters between extreme surges. The bivariate Pareto distribution function *W(x, y)* is

$$W(x, y) = Pr(X < x, Y < y) \tag{8},$$

where $x$ and $y$ represent surges $m$ in Beihai and Dongfang, respectively. Distributions $W_x(x)$ and

$W_y(y)$ are the marginal distributions of $x$ and $y$, respectively. The *CP* distributions are as follows:

$$
\begin{aligned}
CP1 &: Pr(X \geq x \mid Y \geq y) = \frac{Pr(X \geq x, Y \geq y)}{Pr(Y \geq y)} = \frac{1 - W_x(x) - W_y(y) + W(x, y)}{1 - W_y(y)} \\[2mm]
CP2 &: Pr(X \leq x \mid Y \geq y) = \frac{Pr(X \leq x, Y \geq y)}{Pr(Y \geq y)} = \frac{W_x(x) - W(x, y)}{1 - W_y(y)} \\[2mm]
CP3 &: Pr(X \geq x \mid Y \leq y) = \frac{Pr(X \geq x, Y \leq y)}{Pr(Y \leq y)} = \frac{W_y(y) - W(x, y)}{W_y(y)} \\[2mm]
CP4 &: Pr(X \leq x \mid Y \leq y) = \frac{Pr(X \leq x, Y \leq y)}{Pr(Y \leq y)} = \frac{W(x, y)}{W_y(y)}
\end{aligned}
\tag{9}.
$$

The other four CP distributions (Y conditional on X) can be deduced by swapping two variables.

## 4. Extreme-value analysis

160       In this section, we focus on problems in extreme surge prediction that can be solved using the

statistical methodologies of EV analysis. These problems include joint threshold analysis, stochastic

simulation, and the statistics of multivariate extreme surges. Finally, the interpretation of the

statistical results for the extreme surges at the two locations is briefly discussed.

### 4.1 Marginal Transformation and Joint Threshold

165       Generalized Extreme Value Distribution (GEVD) includes EV Ⅰ Ⅱ Ⅲ type distribution and

can describe accurately more EV events than any single component. So we choose that the marginal

distributions of the 2,016 independent events could be described by the following GEVD:

$$F(x) = P(X < x) = exp\left\{ -[1 - \xi(\frac{x - \mu}{\sigma})]^{1/\xi} \right\}, \quad \xi \neq 0 \tag{10},$$

where $\xi$, $\sigma$, and $\mu$ are three GEVD parameters. They are estimated using maximum likelihood,

which was suggested in Coles (2001) and Beirlant et al. (2005). Fig. 3 shows the probability plots

(including the 95% confidence intervals) of the marginal distributions before MGPD is fit and




the estimations of parameters are shown in Tab. 1.

As described in Section 2.1, the MGPD variables must be close to zero in the negative quadrant. The margins of a bivariate distribution can be transformed into uniform ones as suggested by Michel (2007). To standardize the margins, the marginal MGPD distribution must be a negative exponential distribution. Using the Taylor expansion, we can transform the data onto (−1, 0) by computing

$$y = logF(x_i) = log(1 + F(x_i) - 1) \approx F(x_i) - 1 \tag{11},$$

where $F(x_i)$ is the GEVD of index $i$ ($i = 1,2$), indicating Beihai or Dongfang. In Eq. (11), $F(x_i)$ is close to one because this study focuses on extreme observations.

Many dependence models between extreme variables such as logistic, bilogistic, and Dirichlet models have been suggested. However, the choice of dependence model is not usually critical to the accuracy of the final model (Morton and Bowers, 1996). Therefore, a simple bivariate logistic generalized Pareto distribution was selected. The MGPD model of this paper is based on a multivariate EV distribution. Its joint threshold can be calculated by the method from Coles and Tawn (1994). The joint threshold is $c_0 = -0.28$, and there are *218* combinations of Beihai and Dongfang with $c > c_0$. Fig. 4(a) shows the samples that are over the threshold value. In the left subfigure, $c_0 = -0.28$ is a curve, and all observations on the right side of this curve are greater than $c_0$. The converted data is shown in the right subfigure, where $c_0 = -0.28$ is a line.

The correlation parameter $r$ of the dependence function is estimated using maximum likelihood and is $r = 2.15$. Using the obtained estimates for all parameters, the joint extreme probability distribution function is illustrated in Fig. 4(b).

### 4.2 Comparison of Stochastic Simulation Results

Using the simulation method in Section 2.2, we generated a very large sample of bivariate extreme storm surges. In this section, we compare the CP results obtained by simulation and direct calculation. Fig. 5 shows the results of the stochastic simulation for $N = 10,000$ and 100,000. The simulation results are in basic conformity with the observations, which shows that the MGPD simulation method is effective. The scatter diagrams directly show the simulation result; however, they require further quantitative analysis to objectively show their differences.

We use two CPs: *CP1 P(X > x / Y > y)* and *CP4 P(X < x / Y < y)*. Here, *CP1* is the probability





that the surge in Beihai exceeds $x$ conditional on the surge in Dongfang exceeding $y$, and *CP4* is the

probability that the surge in Beihai does not exceed $x$ conditional on the surge in Dongfang not

exceeding y. Fig. 6 shows that the relative difference in value between the simulation and direct

calculation depends on the number of simulations $N$. The subplots in Fig. 6 clearly show that the

relative difference is reduced as $N$ increases. When $N$ reaches $1.5 \times 10^6$, the maximum relative error

between the simulated and calculated results is 8.61%, which we consider satisfactory.

In addition, we conducted runtime experiments in which $1 \times 10^4$, $5 \times 10^4$, $10 \times 10^4$, $1 \times 10^6$, and

$1.5 \times 10^6$ random vectors were generated on a desktop PC with an Intel Core i7 3.4 GHz processor.

The runtimes were 3, 24, 82, 9,649, and 22,106 s.

To estimate the M-year surge of Beihai and Dongfang by means of a univariate analysis, the

Poisson-Gumbel distribution was used (Quek and Cheong, 1992; Naffa et al., 1991), which can be

described by

$$F(x) = P(X < x) = e^{-\lambda[1 - G(x)]}$$
(12),

where $G(x)$ is the Gumbel distribution.

Tables 2 and 3 show the values of *CP1* and *CP4* based on the results from $1.5 \times 10^6$ simulations.

The tables list the calculated and stochastic simulation results for five *CP1* and *CP4* groups for

different combinations of M-year surges at Dongfang and Beihai. The two results are very similar.

For instance, the directly calculated $P(X > x_{50} / Y > y_{10})$ is 12.94% and its simulation result is 14.06%.

Their relative error is 8.61%, which is the maximum relative error for *CP1*. Moreover, the directly

calculated $P(X < x_5 / Y < y_{50})$ is 99.35% and its simulation result is 94.13%. Their relative error is

5.25%, which is the maximum relative error for *CP4*.

### 4.3  Prediction of Extreme Surges

The relationship between extreme surges at Beihai and Dongfang can be analyzed using the

*CP*. Because the peak surges at Dongfang occur earlier than those at Beihai, we use *CP1: P(X > x /*

*Y > y)* to warn of an extreme surge at Beihai. Tab. 2 shows that, when a 50-year surge appears at

Dongfang, the probability of a greater than 50-year surge occurring at Beihai is 94.55%.

Using long-term surge records, we can determine the relationship between extreme surges at



Beihai and Dongfang. Additionally, because of the special geographical relationship between the two locations, a peak surge at Dongfang is a signal that predicts a peak surge at Beihai. We can therefore predict the probability of different surges at Beihai and then take preemptive measures to

reduce or eliminate the associated damages.

## 5. Conclusion and Discussion

### 5.1  Conclusion

The primary aim of this study is to illustrate how recent MGPD developments can be applied to marine disaster prediction. We not only developed a process for determining a joint threshold and

simulation, but we conducted an analysis for extreme surge warnings using MGPD. MGPD is the natural distribution of the MPOT method, which can cull extreme information from raw data. A model based on multivariate EV theory and the intrinsic properties of EVs were also considered. A Monte Carlo MGPD simulation was used to determine the conditional probability of two extreme surges, and the proposed approach was judged to give satisfactory results by comparing the relative

differences in conditional probabilities obtained from observations and simulation data.

### 5.2  New Possibilities Based on Monte Carlo Simulation

Extreme surge warnings would be more reliable if the relationships of extreme surges at three or more sites could be established. Using more related (affected by the same typhoon) sites' storm surge real time data for storm surge warnings of Beihai city, the forecast result may be

more believable by the statistical method presented in the paper. In this paper, the theory of MGPD and its simulation was derived for multidimensional variables. The methodology could be extrapolated to higher dimensional space. Thus, difficulty of solving a procedure for MGPD cannot be exacerbated by the high dimensionality of the variables. A potentially more effective warning approach could be based on a Monte Carlo simulation. Once long-term (e.g., thousands of years)

sea state data has been simulated, several ocean environmental factors can be quickly assessed by the law of large numbers.





## Acknowledgments

This work was supported by the International partnership program of Chinese Academy of Sciences (grant no. 131551KYSB20160002).

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



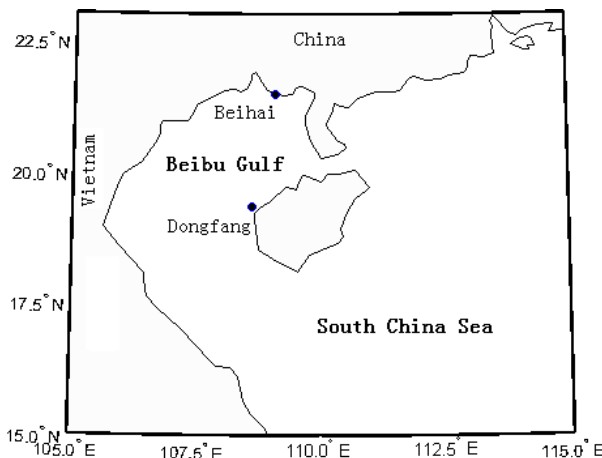

Fig. 1 Location of the two stations in Beibu Bay, SCS

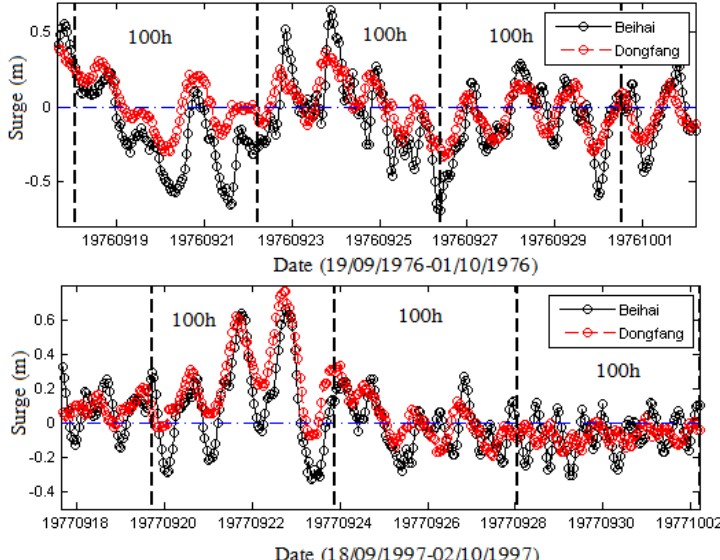


Fig. 2 Declustered surges



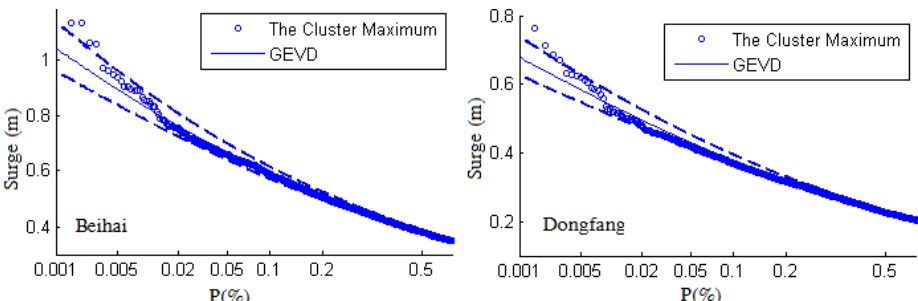

Fig. 3 Fitness testing of the marginal distribution


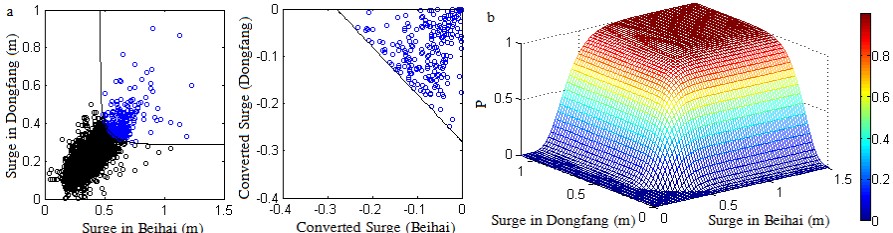

Fig. 4 (a) Dongfang and Beihai observations over the threshold value and (b) joint distribution of EVs for

Dongfang and Beihai






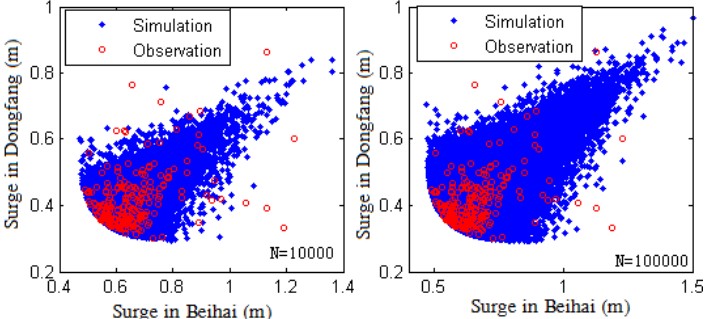

Fig. 5 Values over the threshold and stochastic simulation data


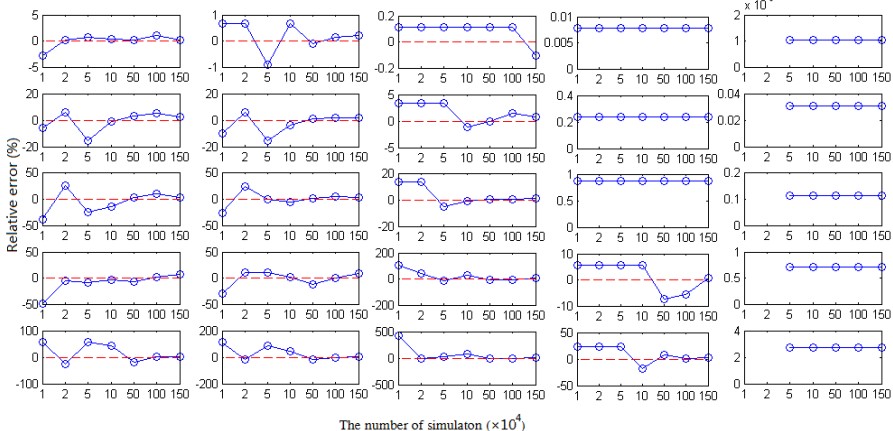

Fig. 6 Variance of the relative error under different numbers of simulation N for *CP1*. The conditional

probabilities for each subplot are shown in Tables 2







Table1 Parameters of the marginal distribution

|          | ξ      | σ      | μ       |
|----------|--------|--------|---------|
| Beihai   | 0.3376 | 0.1187 | −0.0465 |
| Dongfang | 0.1933 | 0.0890 | −0.0720 |


Table 2 Comparison of the results for *CP1 (%)*

with the results from 1.5×10⁶ simulations

| RP (year) |      | 5 | | 10 | | 20 | | 50 | | 100 | |
|-----------|------|---|---|----|---|----|---|----|---|-----|---|
| RP | D(m) B(m) | 0.57 | | 0.77 | | 0.84 | | 0.93 | | 0.99 | |
|    |      | c | s | c | s | c | s | c | s | c | s |
| 5   | 0.87 | 62.52 | 62.59 | 99.33 | 99.54 | 99.89 | 99.79 | 99.99 | 100.00 | 100.00 | 100.00 |
| 10  | 1.13 | 5.20 | 5.34 | 81.00 | 82.52 | 96.67 | 97.44 | 99.76 | 100.00 | 99.97 | 100.00 |
| 20  | 1.22 | 1.70 | 1.74 | 49.38 | 50.71 | 88.05 | 88.70 | 99.13 | 100.00 | 99.89 | 100.00 |
| 50  | 1.34 | 0.34 | 0.37 | 12.94 | 14.06 | 47.95 | 49.68 | 94.55 | 95.12 | 99.29 | 100.00 |
| 100 | 1.42 | 0.11 | 0.11 | 4.29 | 4.47 | 19.09 | 19.40 | 80.75 | 82.93 | 97.37 | 100.00 |

RP: return period, a: analytic solution, s: simulation results, D: Dongfang, B: Beihai. (the same below)







Table 3 Comparison of the results for *CP4 (%)*

370                                     with the results from $1.5 \times 10^6$ simulations

| RP (year) | | 5 | | 10 | | 20 | | 50 | | 100 | |
|---|---|---|---|---|---|---|---|---|---|---|---|
| RP | D(m) B(m) | 0.57 | | 0.77 | | 0.84 | | 0.93 | | 0.99 | |
| | | c | s | c | s | c | s | c | s | c | s |
| 5 | 0.87 | 99.75 | 97.61 | 99.36 | 94.26 | 99.35 | 94.15 | 99.35 | 94.13 | 99.35 | 94.12 |
| 10 | 1.13 | 100.0 | 99.99 | 99.98 | 99.81 | 99.97 | 99.72 | 99.97 | 99.69 | 99.97 | 99.69 |
| 20 | 1.22 | 100.0 | 100.0 | 100.0 | 99.97 | 99.99 | 99.93 | 99.99 | 99.90 | 99.99 | 99.90 |
| 50 | 1.34 | 100.0 | 100.0 | 100.0 | 100.0 | 100.0 | 99.99 | 100.0 | 99.98 | 100.0 | 99.98 |
| 100 | 1.42 | 100.0 | 100.0 | 100.0 | 100.0 | 100.0 | 100.0 | 100.0 | 100.0 | 100.0 | 99.99 |