# Peer review of "Multivariate analysis of extreme storm surges in a semi-enclosed bay"

_Ocean Science, 2016_

## Referee Comment (RC1) · Anonymous Referee #1 · 26 Feb 2017

Review of "Multivariate analysis of extreme storm surges in a semi-enclosed bay".

This paper refers to Beibu Bay, also known as the Gulf of Tonkin, in the South China Sea, where there is data from 2 tide gauge stations around 200 km apart: Dongfang on the island of Hainan and the city of Beihai on the mainland to the north. The premise of the paper is that extreme storm surges at Beihai can by predicted a few hours in advance by surges at Dongfang. The method presented fits a statistical model (a bivariate form of MGPD) to the data at the two sites. A large set of random data is generated according to that model, from which likelihoods are derived, concluding that after removing tidal effects a 50-yr-return surge at Dongfang gives a 95% chance of a 50yr surge at Beihai.

This is a worthwhile goal, and there is a good set of tide data to work on, but I have

major concerns about the methodology used, outlined below. Correcting these would require such substantial reworking that I recommend that the paper is rejected at this time.

There is a fundamental flaw to the paper which must be addressed before any further consideration, which is that the tides are not correctly handled. The surge data used is all non-tidal residuals, and the authors assume that the tides are therefore completely removed. But it is very obvious in Fig 2 that there remains a periodic component. It is possible that this is due to incorrect calculation of tidal harmonic constituents. However it is also possible that there is a nonlinear interaction of surge and tide. For example one effect of the storm surge is to sometimes to alter the timing of the tide, as discussed in eg Pugh  Woodworth's 2014 book, chapter 7. This has the effect of creating an apparently very large surge during the rising tide, but a smaller residual at the high tide. A quick check is to calculate the standard deviation of residuals for different tidal levels. To estimate the magnitude of this effect the authors need to consider the depth of the Bay, described as "shallow", and the tidal phase difference of the sites, but it would be worth checking for existing work in this area, as there may be modelling studies to draw upon.

Either way, the distributions of non-tidal residual surge at Dongfang and Beihai cannot be considered independent, invalidating the statistics used and the conclusion, since the correlation may be due to the tides and not the extreme events - and the extreme events may be exaggerated. A different approach may be required, and at the very least the tides cannot simply be dismissed from the analysis.

Some other concerns - I have not attempted to be comprehensive.

Line 196 is over-optimistic about the fit of the distribution. Fig 3 and 5 show an increasingly poor fit for the most extreme values. From figure 3 in particular I see no justification for extrapolation to 50-yr returns. And the model is only tested on the same data used to generate it.

Line 119 "only" 0.023% missing data... that is pretty low, but do those 2 days correspond to 2 really big storms? If so, how much difference could it make?

The paper could be better structured to aid the reader. It is not obvious early on that the multivariate analysis is used to compare two sites, rather than for example surge and wave variables.

Section 2 is unnecessarily hard to follow, with little explanation of where the sequence of definitions is leading, and a few careless errors. Why does the reader need to know these details? Why do we change from $x_d$ to $x_n$? Does $W_r(x)$ refer to a vector x - if so it would be usual to use bold or underscore. Why the change from X to x? Eq. 4 should be $\frac{\partial^2 W}{\partial x \partial y}$. I assume the transform to "polar" coordinates is what is meant by "converted" in Fig 4... and so on.

It might be better to integrate section 2 to section 4. It may then become apparent which definitions are important and whether key information is missing.

Line 225 reveals that there has been no consideration of the precision of the method or error analysis of the data - I find it extremely unlikely that the probability of a 50-yr surge at the Beihai, given one at Dongfang, can be predicted to 4 significant figures.

A useful addition to 4.3 would be the risk of false-negative - if a warning system were based on extreme surges at Dongfang, what is the likelihood of missing an extreme surge at Beihai?

Fig 1 - a better map would be be useful to the reader, indicating at least the bathymetry, but also eg the major tidal constituents, the dominant typhoon tracks, locations of coastal cities.

All figures could have better captions.

The language is generally good.

---

## Author Comment (AC1) · 2 Mar 2017

Thanks for your review.

In your review, there is a fundamental flaw to my paper, "**which is that the tides are not correctly handled**". The reason is "**The surge data used is all non-tidal residuals, and the authors assume that the tides are therefore completely removed. But it is very obvious in Fig 2 that there remains a periodic component**".

This is a vital problem. If the basic data is wrong, all work will be invalid. From the figure, I can't sure that my method is correct. So I do several tests for proving it.

1 The data of Dongfang and Beihai was analyzed again. The result is the same as before. Like this, carelessness can ruled out.

I used a professional software MIKE for tide analysis and tide prediction, which is an advanced software. 69 harmonic constituents were got.

2 In order to prove my method, I used MIKE analysis a set of tide observation data from Sep 5th 2010 to Sep 5th 2011.
The data have been analyzed by other guys with software Delft3D. And they got 73 tidal harmonic constituents and removed the tides successfully. I use my method in the paper for the data, and the result is close to other guys' that.

[Figure]

Now, I am sure that my method should be feasible. Why does there remain a periodic component in Fig. 2? I try to explain if the phenomenon is reasonable.

On fig.2, there are 14-day (about 350 hours) surge data.

[Figure]

Under the same temporary resolution, the surge from other guys also remains a little periodic component. I think if this phenomenon is even relevant to graphical displays. I didn't get a real explanation. But I can sure that for surge data in the paper, the method should be right.

Other comments in the review, I will rely next time.

---

## Short Comment (SC1) · 5 Mar 2017

It is well structured. The scientific question has been addressed in a appropriate way. It points a new way for statistical forecasts of storm surges, applying new statistical theories and observed data from several adjacent sites.

Language polishing is further needed to make the contents more concise and convincing.

---

## Short Comment (SC2) · 21 Mar 2017

We are pleased that the reviewer sees value in our manuscript. We want to thank the reviewer for his comments.

---

## Short Comment (SC3) · 21 Mar 2017

We thank the Anonymous Referee #1 for the harsh comments and most suggestions which improved the manuscript. But some suggestions is imaginary and unscientific. The words "extremely unlikely" and "invalidating the statistics used and the conclusion" can't accepted completely in an interactive comment on scientific papers.

We can't answer the rest of comments.

---

## Short Comment (SC5) · 22 Mar 2017

The manuscript 'Multivariate analysis of extreme storm surges in a semi-enclosed bay ' by Luo et al. using a multivariate extreme statistical method to predict extreme storm surge probabilities in the Beibu Bay. Being a semi-enclosed bay in the northwestern South China sea, it is an ideal spot for studying the storm surge. The tidal gauge data at two stations are used in the study. The statistical method has its advantages in the probability prediction and can be applied in the area where the observational data can be obtained. The topic of this study is interesting and deserve further investigation. The ms is in good quality and the analyses is reasonable. I recommend this ms being accepted after revision.

There are several aspects need to be addressed during their revision:

[Figure]

1. The description of the methods and its applicability can be improved. The ms need to be reorganized to clarify the methodology and its related physical meaning in the storm surge study. This will lead to a better oceanographic view with the statistic method as a tool.

2. There are only two tidal gauge stations being used in this study. The storm surge usually propagate along the coastline as the typhoon approaching. The comparison is made between simulation and observation at station Beihai in Fig. 5. Which station is used while doing the simulation? Only station Dongfang or both stations? If more stations are available in the future study what can we expected?

3. L137: 'The duration of a typhoon surge in the Beibu Gulf is approximately 100 h'. How to get this specific value of '100 h'? The authors should at least explained how this duration hours are estimated based on how many typhoon events in this area. Will this duration hours affect the final conclusion if someone else get different duration hours?

4. It will be great if the authors can choose a specific period to compare the predicted and observed CP. For example, using the data from 1975 to 1995 to do the statistical analyses and decide the equation, and comparing it with the observed results from 1996-1997. Another question is if the data length can impact the simulation result? This should be indicated in the ms because data samples can be vary at different area.

5.Wording. The ms need to be edited carefully. There are several obviously misspelled and improperly used words in the ms.

---

## Editor Comment (EC1) · J. M. Huthnance (Editor) · 23 Mar 2017

On 22nd March 2017 a short comment by "anonymous" appeared in the discussion of os-2016-94 "Multivariate analysis of extreme storm surges in a semi-enclosed bay". This comment has been removed because only anonymous comments by designated referees are allowed.

---

## Short Comment (SC6) · 26 Mar 2017

sorry, I submited the wrong responss to you. on the other hand, we think get back the (SC3: 'response', Luo Yao, 21 Mar 2017 Âă[reply] ). I regret doing that. you can help me, thanks.

———————————————————

---

## Author Comment (AC2) · 26 Mar 2017

**Authors' response**

We thank Bo Hong for the harsh comments and most suggestions which improved the manuscript.

1. The description of the methods and its applicability are not clear. The ms need to be reorganized to clarify the methodology and its related physical meaning in the storm surge study. This will lead to a better oceanographic view with the statistic method as a tool.

This is a good suggestion. I will describe the processes more compactly. This work is doing, but it can't be finished before deadline of discussion.

2. There are only two tidal gauge stations being used in this study. The storm surge usually propagate along the coastline as the typhoon approaching. The comparison is made between simulation and observation at station Beihai in Fig. 5. Which station is used while doing the simulation? Only station Dongfang or both stations?

For Beibu Gulf, the typhoon usually get to the Beihai from Hailan island, this is a reason why choose the Beibu gulf as the example.
Fig 5 shows the simulation under N=10000 and N=100000 respectively. Two stations is used when doing the simulation.

3. L137: 'The duration of a typhoon surge in the Beibu Gulf is approximately 100 h'. How to get this specific value of '100 h'? The authors should at least explained this duration hours are estimated based on how many typhoon events in this area. Will this duration hours affect the final conclusion if someone else get different duration hours?

In the extreme analysis, the time window for determining block. Actually, this is a difficulties. In lots of papers about extreme analysis (Coles and Tawn, 1994), the time windows weren't determined by factual arguments. So we choosed the time windows by simply analysis in the paper.

(Coles, S. G. and Tawn, J. A.: Statistical methods for multivariate extremes: an application to structural design, J. Roy. Stat. Soc. C-App., 43, 1–48, 1994.)

4. It will be great if the authors can choose a specific period to compare the predicted and observed CP. For example, using the data from 1975 to 1995 to do the statistical analyses, and comparing it with the observed results from 1996-1997. Another question is if the data length can impact the simulation result? This should be indicated in the ms because data samples can be vary at different area.

Yes, data length can impact the simulation. Comparing the result in a specific period is a good method, which can do better for validating the simulation way. We will try to do this.

5.Wording. The ms need to be edited carefully. There are several obviously misspelled and improperly used words in the ms.

Thank you. We will continue polishing the ms. The work is doing always.

---

## Editor Comment (EC2) · J. M. Huthnance (Editor) · 18 Apr 2017

This manuscript has received an anonymous review and two non-referee comments. The referee comments are serious and all comments should be addressed, in the author response and in any revised manuscript. I am making this comment in order to expedite the review process. However, I shall probably want to send any revised manuscript back to the original two designated referees.